# COLLABORATIVE MODELING FOR DOCUMENT-LEVEL EVENT ARGUMENT EXTRACTION

## ABSTRACT

Document-level Event Argument Extraction (EAE) is hampered by two key challenges in long texts: ambiguity among co-occurring events and noise from irrelevant content. To address these issues, we propose CsEAE, a unified framework that comprises two synergistic modules. The co-occurrence-aware module delineates ambiguous event boundaries by modeling dependencies among co-occurring events, while the structure-aware module filters noise by modeling trigger-centric sentence relations. We further extend this framework to Large Language Models (LLMs) with CsLLM, which distills these structural and co-occurrence cues into tailored prompts. Trained on multiple datasets, CsLLM enhances the generalization and performance of LLMs on the EAE task. On the RAMS, WikiEvents, and MLEE benchmarks, CsEAE improves Arg-C F1 scores over the PAIE baseline by 2.1%, 2.3%, and 3.2%, respectively. Our LLM-based approach, CsLLM, achieves even greater performance, demonstrating the effectiveness of our framework.

## 1 INTRODUCTION

Event Argument Extraction (EAE), the task of identifying arguments for specific event roles, aims to extract structured event information from text Peng et al. (2024b;a). As shown in Figure 1, given a trigger, an event type, and a predefined list of roles for that event type, the model is required to extract corresponding text spans as arguments for each role. This structured information can significantly enhance the performance of downstream tasks such as dialogue systems Zhang et al. (2020) and recommendation systems Han et al. (2025).

However, as the length of document-level input texts increases, document-level EAE faces two critical challenges: (1) Ambiguity among co-occurring events He et al. (2023). As illustrated in Figure 1, the four trigger words *crashed*, *stabbed*, *shot*, and *killed* each trigger distinct events. The arguments of these events exhibit an extremely dense distribution, and different events may share identical text spans as arguments for different roles. This dense overlapping obscures the semantic boundaries. (2) Noise from irrelevant content.

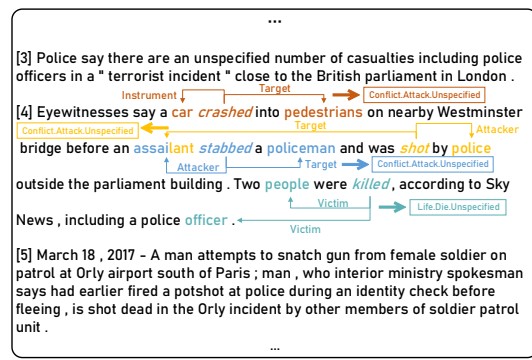

Figure 1: An EAE instance.

The volume of information received by the model increases significantly, including both data useful for extraction and a large amount of irrelevant content that hinders task execution Xu et al. (2022). For example, in sentence [5], person nouns like *man*, *female*, and *soldier* may mislead the model in extracting the *Victim* role for the *Life.Die.Unspecified* event triggered by *killed*. Notably, prior work has failed to address both challenges simultaneously Ma et al. (2022); He et al. (2023); Liu et al. (2024).

To address these challenges simultaneously, we propose CsEAE, a co-occurrence-aware and structure-aware framework for EAE. The core of CsEAE employs two synergistic modules to help the model capture event boundaries and focus on critical information. 1. Co-occurrence-aware module that captures interactions among co-occurring events to delineate semantic boundaries, by ex-

plicitly marking all triggers and encoding relevant templates. 2. Structure-aware module that filters redundant information by focusing on trigger-centric content. We observe a high locality of event information: over 94% (WikiEvents), 82% (RAMS), and 99% (MLEE) of arguments reside in the same sentence as triggers. This underscores the importance of the trigger's sentence. Consequently, this module constructs trigger-centric sentence-level relation to guide the model to selectively attend to the trigger sentence and its most relevant neighboring sentences, while reducing irrelevant noise.

Building on the design of CsEAE, we extend this framework to LLMs by introducing CsLLM. This method encodes the core strategies validated in CsEAE, namely co-occurrence awareness and structure awareness, into the prompt space. Rather than altering the LLM's architecture, CsLLM guides its reasoning by encoding co-occurrence signals and reinforcing a structural focus within the prompt itself. This prompt-based distillation, combined with a multi-dataset fine-tuning strategy to enhance generalization, showcases an effective path for applying complex EAE strategies to the LLM paradigm. Our contributions are summarized as follows:

- We propose CsEAE, a framework with co-occurrence-aware and structure-aware modules that for the first time provides a unified solution to the intertwined challenges of event boundary ambiguity and information redundancy.

- We introduce CsLLM, which demonstrates that the core strategies from CsEAE can be effectively distilled into prompts to steer LLMs. This approach, combined with multi-dataset training, significantly improves the performance and generalization of LLMs on the document-level EAE task and offers a novel, architecture-agnostic path for complex EAE tasks.

- Experiments on the RAMS, WikiEvents, and MLEE benchmarks show that CsEAE improves upon the PAIE baseline by 2.1%, 2.3%, and 3.2% in Arg-C F1 scores, respectively. CsLLM also achieves superior performance, further validating the effectiveness of our core strategies.

## 2 CsEAE Model

### 2.1 Model Architecture Overview

CsEAE's architecture processes the input document $\mathcal{D}$ in two primary stages.

**Stage 1: Event-Oriented Context Representation.** This stage, with its data flow depicted by the green arrows in Figure 2, generates the event-oriented context representation $H_{\mathcal{D}}$, which is designed to be aware of sentence structure and event co-occurrence. We first use a structure-aware encoder $\text{Encoder}_{Sap}$ on the input $\mathcal{D}$ to produce an initial representation $H_{\mathcal{D}}^{enc}$. This is then passed to a co-occurrence-aware decoder $\text{Decoder}_{Cap}$ to generate the final $H_{\mathcal{D}}$.

$$
\begin{aligned}
H_{\mathcal{D}}^{enc} &= \text{Encoder}_{Sap}(\mathcal{D}), \\
H_{\mathcal{D}} &= \text{Decoder}_{Cap}(H_{\mathcal{D}}^{enc}, H_{\mathcal{D}}^{enc}).
\end{aligned}
\tag{1}
$$

**Stage 2: Context-Oriented Prompt Representation.** This stage, following the orange arrows, generates the context-oriented template representation $H_{pt}$, a specialized query for the target event role. A structure-aware decoder $\text{Decoder}_{Sap}$ produces this representation by taking the event-type template $p_{e_n}$ as its input query, while using the document representation $H_{\mathcal{D}}^{enc}$ from Stage 1 as its cross-attention context.

$$
H_{pt} = \text{Decoder}_{Sap}(H_{\mathcal{D}}^{enc}, p_{e_n}).
\tag{2}
$$

Finally, $H_{\mathcal{D}}$ and $H_{pt}$ are jointly fed into the final span selection module, as detailed in Section 2.4. The specific implementations of our structure-aware and co-occurrence-aware mechanisms are detailed in Sections 2.2 and 2.3.

### 2.2 Co-occurrence-Aware Module

The co-occurrence-aware module is designed to model dependencies among co-occurring events. It consists of three key components: context labeling, co-occurring template encoding, and prefix generation.

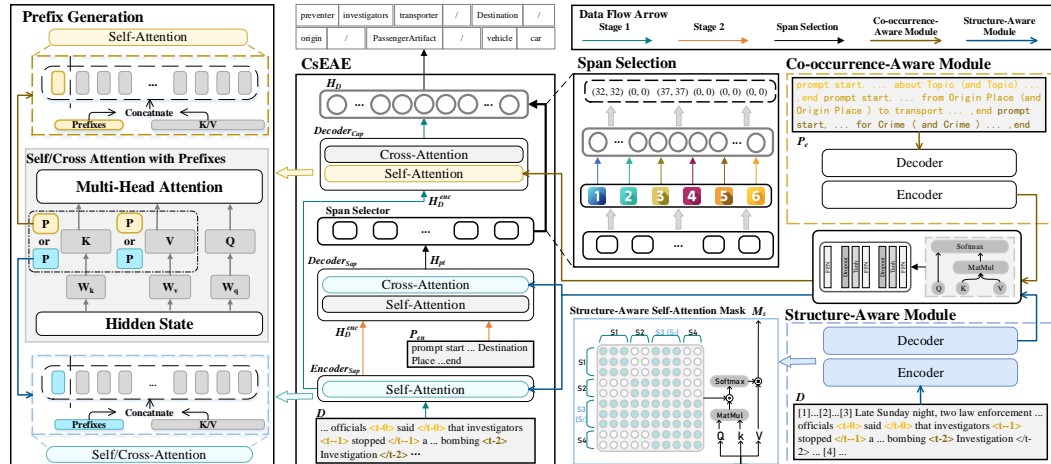

Figure 2: An overview of CsEAE, which enhances a standard encoder-decoder backbone with two synergistic modules. The co-occurrence-aware module leverages templates of co-occurring events to generate prefixes, creating the Decoder$_{Cap}$. The structure-aware module uses a structure-aware self attention mask over the document to generate another set of prefixes, producing the Encoder$_{Sap}$ and Decoder$_{Sap}$. The prefix generation mechanism serves to inject this specialized knowledge into the backbone. These customized components are then utilized to generate two key representations in two stages:$H_{\mathcal{D}}$ and $H_{pt}$. Finally, both representations are fed into the span selection module for argument extraction.

### 2.2.1 CONTEXT LABELING

Given the input of the model $\mathcal{D} = \{t_1, t_2, \ldots, t_n\}$, where $t_i$ represents the $i$-th token in the input. Given $E = \{e_0, e_1, \ldots, e_l\}$, where $e_i$ represents one event appearing in $\mathcal{D}$, and $l$ represents the number of events appearing in $\mathcal{D}$. Given all the triggers $T = \{e_0^t, e_1^t, \ldots, e_l^t\}$, where $e_i^t$ represents the trigger corresponding to event $e_i$, and $e_i^t$ corresponds one-to-one with $e_i$. We annotate all token spans corresponding to triggers in $\mathcal{D}$ according to the order in which the triggers $e_i^t$ appear in $\mathcal{D}$. Specifically, for the trigger $e_n^t$ corresponding to the event $e_n$ being extracted, we will annotate its appearance in $\mathcal{D}$ using special characters <t- -1>and </t- -1>.

For triggers $e_j^t$ corresponding to other events existing in $\mathcal{D}$, we will annotate them according to the order of appearance in $\mathcal{D}$ using <t-$k$>and </t-$k$>, where $k$ is calculated starting from 0 and incremented by 1.

### 2.2.2 CO-OCCURRING TEMPLATE ENCODING

Given $P_e = \{p_{e_1}, p_{e_2}, \ldots, p_{e_l}\}$, where $p_{e_i}$ represents the template corresponding to event $e_i$. As $p_{e_i}$, $e_i^t$, and $e_i$ are uniquely paired. In this paper, we utilize templates proposed in PAIE Ma et al. (2022) for the RAMS and WikiEvents datasets and TabEAE He et al. (2023) for the MLEE dataset. To fully utilize the semantic information provided by the templates, we first concatenate all templates $P_e$ corresponding to events mentioned in $\mathcal{D}$. Then, we encode them into the backbone to obtain dense vector representations $W_C$ for all co-occurring event templates. Finally, the information of $W_C$ is integrated into the prefixes.

### 2.2.3 PREFIX GENERATION

After constructing the co-occurrence-aware matrix $W_C$, we condense $W_C$ into a set of prefixes Li & Liang (2021); Hsu et al. (2023b). These prefixes are then injected into the backbone's decoder to make it co-occurrence-aware, a process visualized by the yellow arrows in Figure 2. Firstly, we define a learnable vector of length $len$, which serves as the Q vector for multi-head attention, where $len$ is a tunable hyperparameter controlling the final length of the prefixes to be fed into the backbone, we set it as 40. Then, $W_C$ is used as the K and V vectors in multi-head attention computation, which is computed with the Q vector. After multi-head attention computation, we

obtain a set of compressed dense vector $\mathcal{P}$, which then undergoes a series of linear layers. Finally, $\mathcal{P}$ is evenly split into $c$ segments $\mathcal{P} = \{\mathcal{P}_1, \mathcal{P}_2, \ldots, \mathcal{P}_c\}$, each with a length of $len$, where $c$ is the number of transformer layers in the backbone. This results in prefixes that can be concatenated into the backbone for computation, and the decoder augmented with these co-occurrence prefixes is thus denoted as $\text{Decoder}_{Cap}$.

## 2.3 STRUCTURE-AWARE MODULE

The structure-aware module aims to filter irrelevant noise by focusing on trigger-centric content. It achieves this through two mechanisms: structural relationship and structure prefixes.

### 2.3.1 STRUCTURAL RELATIONSHIP

For different document inputs $\mathcal{D}$, as shown in Figure2, we designed a structure-aware self-attention mask $M_s$, which treats sentences as units and trains the model to be structure-aware across the entire document. Specifically, given the document-level input $\mathcal{D} = \{S_1, S_2, \ldots, S_m\}$, where $S_i$ represents the $i$-th sentence in $\mathcal{D}$, and given the trigger $e_n^t$ of the current event to be extracted, located in sentence $S_n$, $M_s$ restricts the receptive field of all sentences except $S_n$, allowing these sentences to focus only on themselves and $S_n$. In contrast, $S_n$ can attend to all sentences.

We can obtain the structure-aware dense vector representation $W_S$ for the inputs $\mathcal{D}$ as follows:

$$W_S = Decoder(Encoder(\mathcal{D}, M_s)). \tag{3}$$

Finally, following the same procedure described in Section 2.2.3, $W_S$ is condensed into a set of structure-aware prefixes. As depicted by the blue arrows in Figure 2, these prefixes are injected into the backbone's standard encoder and decoder, forming the $\text{Encoder}_{Sap}$ and $\text{Decoder}_{Sap}$.

## 2.4 SPAN SELECTION

After obtaining $H_{pt}$, we extract the slot representation $\psi_k$ corresponding to the pre-defined roles from $H_{pt}$, where $k$ represents the $k$-th slot. Then, we convert $\psi_k$ into a span selector specific to that slot $\theta_k$ Ma et al. (2022); Du & Cardie (2020b). Next, apply the span selector $\theta_k$ directly to the event-oriented context representation $H_{\mathcal{D}}$ to determine the argument's token span $[p_k^{(start)}; p_k^{(end)}]$.

$$
\begin{aligned}
\psi_k^{(start)} &= \psi_k \circ w^{(start)} \in R^h, \\
\psi_k^{(end)} &= \psi_k \circ w^{(end)} \in R^h, \\
\text{logit}_k^{(start)} &= \psi_k^{(start)} H_{\mathcal{D}} \in R^L, \\
\text{logit}_k^{(end)} &= \psi_k^{(end)} H_{\mathcal{D}} \in R^L, \\
p_k^{(start)} &= \text{Softmax}(\text{logit}_k^{(start)}) \in R^L, \\
p_k^{(end)} &= \text{Softmax}(\text{logit}_k^{(end)}) \in R^L.
\end{aligned}
\tag{4}
$$

Where $\theta = [w^{(start)}; w^{(end)}] \in R^{h \times 2}$ is a learnable parameter matrix shared by all span selectors, $\circ$ represents element-wise multiplication. $\theta_k = [\psi_k^{(start)}; \psi_k^{(end)}]$ is the span selector specific to the slot corresponding to the role, $L$ denotes the context length. We define the loss function $\mathcal{L}$ as:

$$
\begin{aligned}
\mathcal{L}_k(\mathcal{D}) &= -(\log p_k^{(start)}(s_k) + \log p_k^{(end)}(e_k)), \\
\mathcal{L} &= \sum_{\mathcal{D} \in B} \sum_k \mathcal{L}_k(\mathcal{D}).
\end{aligned}
\tag{5}
$$

Where $B$ ranges over all context in dataset and $k$ ranges over all slots in template $p_{e_n}$ for $\mathcal{D}$, and $(s_k, e_k)$ represents the token span of the most likely argument corresponding to the role in $H_{\mathcal{D}}$.

During the inference phase, we predefine spans $\mathcal{C}$ that cover all possible spans within a predefined length and include a special span (0, 0) to represent the absence of any corresponding argument. Then, we utilize the span selector $\theta_k$ to compute scores for all spans using the following method:

$$\text{score}_k(i, j) = \text{logit}_k^{(start)}(i) + \text{logit}_k^{(end)}(j). \tag{6}$$

Where $i$ and $j$ represent the start and end indices of each span in the set of spans.

Based on the scores, we determine the predicted final span by selecting the span with the highest score: $(\widehat{s_k}, \widehat{e_k}) = \arg\max_{(i,j) \in \mathcal{C}} \text{score}_k(i,j)$.

For the issue of multiple arguments of the same role, we utilize the Hungarian algorithm Kuhn (1955). For the problem of allocating multiple slots corresponding to a single role, we employ Bipartite Matching Carion et al. (2020); Yang et al. (2021).

# 3 CSLLM: LLM ENHANCEMENT

## 3.1 PROMPT DESIGN

Given the input $\mathcal{D}$, we designed a corresponding prompt $\mathcal{P}_\mathcal{L}(\mathcal{D})$ for LLMs. As shown in Figure3, the prompt $\mathcal{P}_\mathcal{L}(\mathcal{D})$ is divided into three parts:

$$\mathcal{P}_\mathcal{L}(\mathcal{D}) = [\mathcal{I}; \mathcal{E}; \mathcal{Q}]. \tag{7}$$

The first part is the instruction $\mathcal{I}$, which describes the task and provides basic information such as the trigger, roles, and output format. The second part is the example $\mathcal{E}$, which provides a single example to the LLMs. We identified corresponding examples for each event type from the training set and the example should include as many arguments as possible from the input. The third part is the question $\mathcal{Q}$. We use <doc>for input to separate the $\mathcal{Q}$ from other components in the prompt.

You will perform event argument extraction tasks in the news domain. Please follow the steps below to identify the arguments corresponding to the given roles in the document marked by <doc>. If a role does not have a corresponding argument, strictly output None. In step 4, I will provide you with an example marked by <eg>.

1 – The trigger word 'explosion' marked with <t> triggers a Conflict.Attack.DetonateExplode event and all trigger words that trigger other events are marked by <T>. Additionally, you need to pay close attention to the sentence marked by  in the document.
2 – The event 'Conflict.Attack.DetonateExplode' corresponds to the list of roles: Attacker, Target, Instrument, ExplosiveDevice, Place.
3 – Please output the role names and their corresponding arguments in JSON format.
4 – I will give you an example as follows:
<eg> Given a document: ...
Denis Broüquier , the city ' s district mayor , told press that " the charge was too small to kill , " and a government source told AFP news agency it had been a " relatively weak explosive charge " ...
You need to output: {"Attacker": "Jihadist", "ExplosiveDevice": "bomb", "Instrument": "gun", "Target": "people", "Place": "France"} <eg>.

Document: <doc> ... a senior defense official said .  " We have information that 126 people have been <T> killed <T> in the <t> explosion <t> inside the military training center , eight special commandoes are among the dead , " said a senior official in the defense ministry in Kabul , speaking on condition of anonymity .  The official said the assault began on Monday morning when the attackers rammed a car full ... <doc>

Figure 3: The blue parts represent $\mathcal{I}$, the yellow parts represent $\mathcal{E}$, the green parts represent $\mathcal{Q}$ and the red parts represent co-occurrence-aware and structure-aware interactions.

## 3.2 STRATEGY-DRIVEN PROMPT ENHANCEMENT

The standard prompt design provides a foundation. To address the core challenges of ambiguity and noise within the LLM paradigm, we enhance this prompt by injecting the co-occurrence and structure-aware principles validated in CsEAE. This approach translates our framework's core logic from the parameter space of smaller models to the prompt space of LLMs.

As highlighted in red in Figure 3, these enhancements are implemented through targeted markings. We introduce co-occurrence awareness by explicitly annotating all event triggers within the input context $\mathcal{Q}$. This provides the LLM with a clear map of all co-occurring events. Structure awareness is similarly encoded by marking the entire sentence that contains the target trigger. These markings are complemented by modifications to the instruction $\mathcal{I}$, which now guides the LLM to focus its reasoning on these highlighted triggers and sentences.

This strategy offers a flexible, architecture-agnostic path for enhancing LLMs on complex EAE tasks while fully leveraging their in-context learning capabilities. To further bolster the performance and generalization of our final model, CsLLM, we also adopted a multi-dataset fine-tuning approach, training the model on a composite dataset to expose it to a wider range of event patterns.

# 4 EXPERIMENTS

## 4.1 DATASETS AND IMPLEMENTATION DETAILS

We used the three most commonly employed datasets for document-level EAE: RAMS Ebner et al. (2020), WikiEvents Li et al. (2021), and MLEE Pyysalo et al. (2012). To further enhance model training, we also incorporated sentence-level EAE datasets, specifically ACE Doddington et al. (2004) and GENEVA Parekh et al. (2023). Additionally, to more comprehensively validate the effectiveness of CsEAE, we applied the data processing methods used in TextEE Huang et al. (2024)

to WikiEvents and RAMS. These methods included standardization of data assumptions, normalization of data processing steps, and standardization of 5 times dataset splits. We leave the dataset details in Appendix B, and the implementation details in Appendix C.

## 4.2 BASELINES

We compare CsEAE and CsLLM against the following two types of baselines: **(1) PLM-based methods**: EEQA Du & Cardie (2020a), TSAR Xu et al. (2022), TagPrime-C/CR Hsu et al. (2023a), Bart-Gen Li et al. (2021), PAIE Ma et al. (2022), TabEAE He et al. (2023), DEEIA Liu et al. (2024), Fusion Ding et al. (2025), HMPEAE Zhang et al. (2024), HD-LoA Zhou et al. (2024). **(2) LLM-based methods**: Chat-GPT, GPT4o, GPT4o-mini [1] (In-Context Learning), and Llama3-8B, Llama3-8B-Instruct Touvron et al. (2023) (SFT).

## 4.3 EVALUATION METRICS

Following previous works Ma et al. (2022), we used Arg-I F1 and Arg-C F1 metrics to evaluate performance on argument identification and classification. Note that in all experiments, Arg-I and Arg-C are equivalent to Arg-I+ and Arg-C+ as defined in TextEE. More details in Appendix D.

## 4.4 MAIN RESULTS

### 4.4.1 CSEAE

We evaluate the proposed model CsEAE and baseline methods under all benchmarks. In Table1, our model outperformed all baselines on all datasets. Compared to the baseline PAIE Ma et al. (2022), CsEAE achieves improvements on the RAMS dataset, with increases of 2.2% in Arg-I and 2.1% in Arg-C F1 scores, respectively. On the WikiEvents dataset, CsEAE shows improvements of 2.0% in Arg-I and 2.3% in Arg-C metrics. Similarly, on the MLEE dataset, CsEAE achieves improvements of 3.0% in Arg-I and 3.2% in Arg-C metrics. The consistent improvements of 2% or more across all datasets demonstrate the effectiveness of the structure-aware and co-occurrence-aware modules.

Table 1: Overall performance of various models. All experiments utilized a large-scale backbone. The highest scores are colored red, and the second-highest are colored orange. For prompt-reliant methods, we did not replicate dataset metrics (marked with * in table) when original publications lacked the specific prompts.

| Model | RAMS | | WikiEvents | | MLEE | |
|---|---|---|---|---|---|---|
| | Arg-I | Arg-C | Arg-I | Arg-C | Arg-I | Arg-C |
| EEQA | 50.2 | 46.8 | 59.7 | 55.4 | 69.1 | 66.9 |
| TSAR | 56.2 | 50.9 | 69.8 | 64.4 | 71.9 | 70.1 |
| BART-Gen | 51.6 | 48.2 | 65.4 | 60.9 | 71.1 | 69.2 |
| DEEIA | 55.9 | 51.3 | 68.2 | 63.0 | 73.5 | 72.5 |
| TabEAE-m2s | 56.2 | 51.4 | 69.7 | 64.9 | - | - |
| TabEAE-m2m | 55.9 | 50.9 | 70.3 | 64.6 | 74.0 | 72.9 |
| PAIE | 55.3 | 51.0 | 68.9 | 64.2 | 71.3 | 70.1 |
| Fusion* | 48.8 | 43.1 | 60.6 | 55.5 | - | - |
| HMPEAE* | 57.0 | 52.6 | 69.7 | 63.7 | - | - |
| HD-LoA* | 52.1 | 46.9 | - | - | - | - |
| **CsEAE** | 57.5 | 53.1 | 70.9 | 66.5 | 74.3 | 73.3 |

We also utilized the data preprocessing method provided by TextEE. The final results, shown in Table 2, represent the average performance across these five splits. Even under such uniform experimental conditions with robustness validation, CsEAE consistently outperforms all baselines on Arg-C, demonstrating its superior effectiveness.

### 4.4.2 CSLLM

---

[1] https://openai.com

As shown in Table 3, further improvements were achieved when the model was fine-tuned on WikiEvents, RAMS and MLEE, demonstrating that LLMs can effectively leverage their robust memory capacity to learn generalizable extraction capabilities across diverse sources. Incorporating two additional sentence-level datasets further boosted performance. Moreover, introducing co-occurrence-aware and structure-aware module into the prompts led to additional gains over models fine-

Table 2: All experiments in table below used the data processing methods described in TextEE, and the results are averaged over five data splits. * means the value from the TextEE's paper.

| Model | RAMS | | WikiEvents | |
|---|---|---|---|---|
| | Arg-I | Arg-C | Arg-I | Arg-C |
| TagPrime-C* | 54.4 | 48.3 | 68.6 | 64.0 |
| TagPrime-CR* | 54.1 | 49.7 | 68.4 | 65.5 |
| EEQA* | 48.9 | 44.7 | 48.4 | 46.1 |
| BART-Gen* | 50.4 | 45.4 | 68.1 | 63.9 |
| PAIE* | 55.2 | 50.5 | 69.8 | 65.2 |
| **CsEAE** | 56.8 | 52.3 | 69.3 | 65.7 |

tuned on single datasets without such enhancements, suggesting that insights effective in smaller models also benefit LLMs. However, CsLLM (ALL) performed worse on RAMS than CsEAE, likely due to the limited structural encoding in prompts. Unlike CsEAE, which integrates structure-aware mechanisms directly, LLMs rely on implicit prompt-based guidance. This soft constraint has two limitations: 1. the model may over-attend to high-frequency or superficial features due to pretraining biases, and 2. structural relationships are hard to encode effectively via prompts.

For further experimental analysis under the ICL setting, please see Appendix F. We also present additional generalization experiments in Appendix G.

Table 3: Overall performance of LLMs. **Doc** represents training using the WikiEvents, RAMS, and MLEE. **ALL** signifies that all five datasets were used. CsLLM used Llama3-Instruct as the LLMs.

| Model | WikiEvents | | RAMS | | MLEE | |
|---|---|---|---|---|---|---|
| | Arg-I | Arg-C | Arg-I | Arg-C | Arg-I | Arg-C |
| *In-context Learning* | | | | | | |
| GPT-3.5 | 18.12 | 16.04 | 34.30 | 27.64 | 21.16 | 15.46 |
| GPT4o-mini | 20.42 | 17.99 | 35.47 | 30.04 | 25.85 | 22.34 |
| GPT4o | 25.58 | 23.37 | 41.58 | 35.70 | 28.04 | 24.92 |
| *Fine-tuning* | | | | | | |
| Llama3 | 65.82 | 60.68 | 37.00 | 33.26 | 72.63 | 71.09 |
| Llama3-Instruct | 65.88 | 60.54 | 55.06 | 49.82 | 70.85 | 69.76 |
| **CsLLM** | 66.33 | 62.80 | 55.35 | 50.25 | 74.80 | 73.87 |
| **CsLLM (Doc)** | 69.92 | 65.66 | 56.14 | 50.99 | 75.34 | 74.10 |
| **CsLLM (ALL)** | 70.89 | 66.53 | 57.19 | 51.84 | 75.93 | 74.89 |

## 5 ANALYSIS

### 5.1 ABLATION STUDIES

Table 4: Ablation study on all benchmarks, str: structure-aware, occur: co-occurrence-aware.

| Model | RAMS | | WikiEvents | | MLEE | |
|---|---|---|---|---|---|---|
| | Arg-I | Arg-C | Arg-I | Arg-C | Arg-I | Arg-C |
| w/o str&occur | 55.3 | 51.0 | 68.9 | 64.2 | 71.3 | 70.1 |
| add str | 55.8 | 52.0 | 70.5 | 64.8 | 72.0 | 70.9 |
| add occur | 55.9 | 51.6 | 70.5 | 65.9 | 73.9 | 72.9 |
| **CsEAE** | 57.5 | 53.1 | 70.9 | 66.5 | 74.3 | 73.3 |

As shown in Table 4, both structure-aware and co-occurrence-aware module independently improve performance across all datasets. The structure-aware module significantly boosts Arg-C on the RAMS dataset (+1.0%), likely due to its stable sentence structure, with each document consisting of five sentences. In contrast, the co-occurrence-aware module brings greater improvements on

the WikiEvents and MLEE datasets (Arg-C increases of +1.7% and +2.8%, respectively), attributed to the higher event density and structural complexity. By integrating both interaction mechanisms, CsEAE achieves the best overall performance, with average improvements of 2.4% in Arg-I and 2.53% in Arg-C.

## 5.2 Capturing the Event Semantic Boundary

Following TabEAE, we analyzed CsEAE's ability to capture event semantic boundaries on the WikiEvents and MLEE datasets. These datasets were chosen due to their larger number of events and more complex event relationships. Our analysis was conducted from two perspectives: inter-event and intra-event semantics.

**Inter-event semantics.** We categorized dataset instances by event overlap (shared argument spans) and non-overlap (N_O). As Figure 4 shows, CsEAE improved overall across both datasets, excelling particularly with overlap instances. For example, on WikiEvents' overlap instances, CsEAE's Arg-C performance improved by 5.4% over TabEAE and 3.4% over PAIE.

**Inner-event semantics.** We categorized argument roles by their distance from the trigger, defined as the maximum head word index difference (argument - trigger) for all arguments of a role $d$. Figure 4 illustrates this, with negative values indicating arguments left of the trigger. CsEAE achieved superior performance across various distance ranges on both datasets, showing a trend of increasing improvement with greater distances. For instance, on the WikiEvents dataset, when the distance $d \geq 15$, CsEAE's Arg-C performance improved by 14.0% over TabEAE and 6.9% over PAIE, confirming its strong ability to capture event semantics. Additionally, in Appendix E, we demonstrate the model's improved robustness to noise.

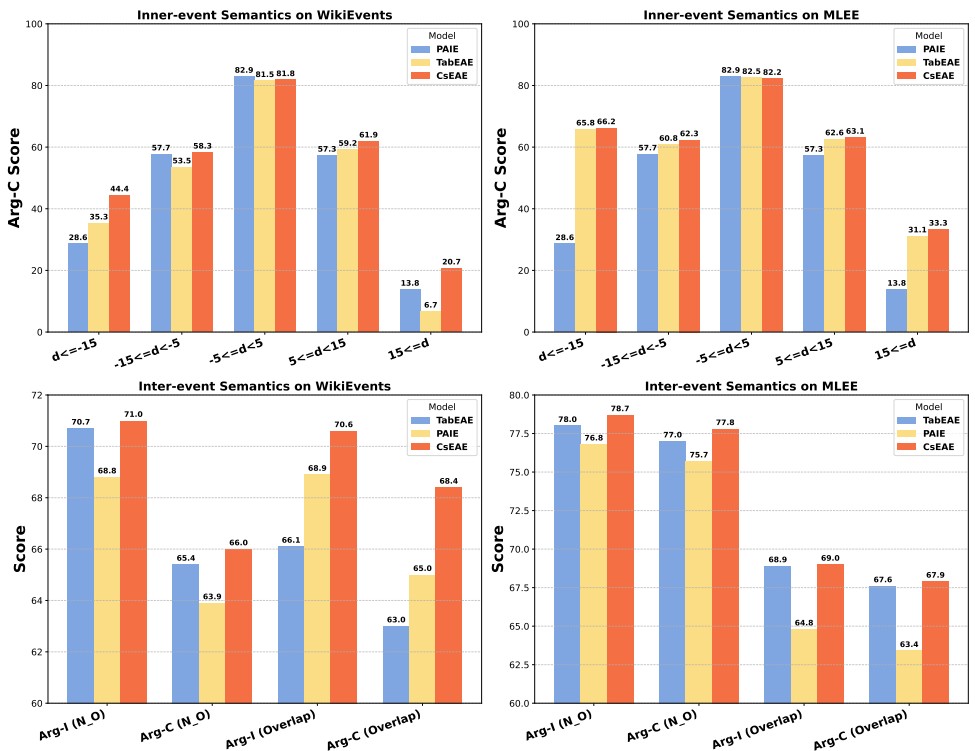

Figure 4: Detailed performance analysis on inner-event and inter-event semantics.

## 5.3 Transfer Capability

To further evaluate the model's transferability, we conducted a one-shot transfer experiment, with detailed results in Table 5. In this setup, the model is evaluated directly on unseen datasets. The results show that after multi-dataset fine-tuning, the model exhibits exceptional generalization capabilities, with its performance on MLEE even surpassing that of GPT-4o. The model trained on

DOC$_{\text{w/o MLEE}}$ achieved the best performance, confirming the effectiveness of multi-dataset training in enhancing transferability. This suggests that: 1. multi-dataset fine-tuning promotes a deeper understanding of core patterns in event argument extraction; and 2. the learning process transcends memorizing dataset-specific features in favor of abstracting generalizable rules.

Notably, when sentence-level dataset were added to training data, transfer performance declined. We initially hypothesized that diverse datasets could enhance performance in data-scarce scenarios despite domain or structural mismatches. Yet results revealed challenges of heterogeneity: differing annotation schemes and granularities caused negative transfer, weakening generalization to unseen

Table 5: One-shot transfer experiments.

| Model | MLEE | | GENEVA | |
|---|---|---|---|---|
| | Arg-I | Arg-C | Arg-I | Arg-C |
| GPT4o | 28.04 | 24.92 | 42.98 | 39.55 |
| WikiEvents | 34.42 | 30.03 | 29.29 | 26.69 |
| DOC$_{\text{w/o MLEE}}$ | 39.06 | 36.84 | 29.24 | 27.11 |
| DOC | - | - | 16.75 | 15.61 |
| ALL$_{\text{w/o MLEE}}$ | 18.76 | 17.70 | - | - |

domains. Our findings suggest that positive transfer in one-shot settings requires auxiliary and target data to be homogeneous in domain and structure, such as both at document- or sentence-level.

On the GENEVA dataset, which involves fine-grained event classification, GPT-4o showed clear superiority. This likely stems from GENEVA's many semantically similar, easily confusable event types, which GPT-4o's scale and knowledge help distinguish more effectively.

### 5.4 ZERO-SHOT OF LLMs

To evaluate the model's zero-shot capability, we removed the example part $\mathcal{E}$ from the prompt. As shown in Figure 5, we found that under multi-dataset fine-tuning, performance on the RAMS and WikiEvents datasets degraded as more training data was added, exhibiting negative transfer. This stands in sharp contrast to the results in the one-shot setting. However, the domain-distinct MLEE dataset

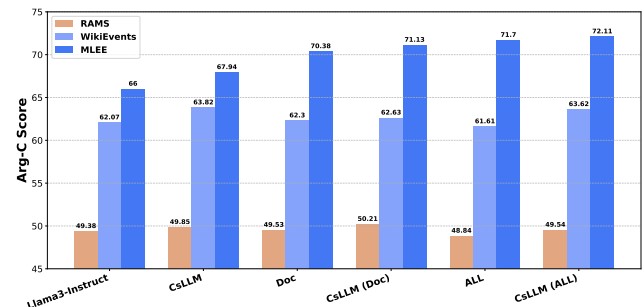

Figure 5: The zero-shot performance

maintained its performance gains. We attribute this to domain confusion. In the zero-shot setting, the model lacks the demonstrations needed to distinguish between domains. When the prompt for different datasets are identical (e.g., RAMS and WikiEvents), the model experiences representational conflict. In contrast, MLEE's prompt template contains unique medical terms that act as implicit anchors, helping the model differentiate domain features and thereby avoiding negative transfer. In Appendix H, we conduct experiments on transfer capabilities in a zero-shot setting.

## 6 CONCLUSION

In this paper, we addressed two critical and intertwined challenges in document-level Event Argument Extraction (EAE): the semantic ambiguity arising from co-occurring events and the information noise from irrelevant content. We proposed CsEAE, a unified framework featuring two synergistic modules. The co-occurrence-aware module helps the model capture event boundaries by modeling interactions among co-occurring events, while the structure-aware module effectively filters noise by focusing on trigger-centric sentence relations. Furthermore, we extended our framework's core principles to the LLM paradigm by introducing CsLLM. This approach innovatively demonstrates that fine-grained strategies, validated in smaller models, can be effectively distilled into the prompt space to steer powerful LLMs. Our experiments on the RAMS, WikiEvents, and MLEE benchmarks confirm the effectiveness of our methods, with both CsEAE and CsLLM achieving significant performance gains.

## REPRODUCIBILITY STATEMENT

To ensure the reproducibility of our research, we provide detailed information regarding our code, experimental setup, and computational environment. Our source code, which includes the implementations for both CsEAE and CsLLM, is available in the supplementary materials. For a comprehensive list of hyperparameters, library versions, and specific training procedures for all models, please refer to our implementation details in Appendix C. The appendix also specifies the hardware used for our experiments.

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

## A  RELATED WORKS

### A.1  DOCUMENT-LEVEL EVENT ARGUMENT EXTRACTION

Document-level EAE, capable of extracting events across multiple sentences, has attracted increasing attention. Some works leverage abstract meaning representation for this task Xu et al. (2022); Yang et al. (2023). PAIE Ma et al. (2022) enhances this with manually designed slot prompts.

TabEAE He et al. (2023) formulates EAE as a table-filling task, enabling simultaneous extraction of all events. HMPEAE Zhang et al. (2024) mitigates intra-class variance by representing roles with multiple prototypes on a hypersphere. Fusion-based methods Ding et al. (2025) integrate selective and generative approaches using fusion prompts and a unified learning strategy.

### A.2 LLMs for Event Argument Extraction

To enhance LLM performance in information extraction, researchers have proposed various innovative methods. LLM-IE Ma et al. (2023) adopts a "filter-then-rerank" paradigm, where a fine-tuned smaller model filters initial predictions, and an LLM reranks challenging cases. For data augmentation, STAR Ma et al. (2024) uses a "structure-to-text" strategy that generates data structures first, then passages, refining them through self-reflection to improve low-resource performance. Another approach Chen et al. (2024) treats LLMs as expert annotators, generating data aligned with benchmark distributions by incorporating training samples into prompts. To optimize prompting, HD-LoA Zhou et al. (2024) formalizes example selection using task heuristics and analogy-based prompting, helping LLMs generalize to new scenarios. GoLLIE Sainz et al. (2024), meanwhile, fine-tunes LLMs on human annotation guidelines, enabling them to learn and follow complex rules.

## B DATASETS

Table 6: Basic information for the datasets used. "Args" stands for Arguments.

| Statistic | RAMS | WikiEvents | MLEE |
|---|---|---|---|
| Event types | 139 | 50 | 23 |
| Args per event | 2.33 | 1.40 | 1.29 |
| Events per text | 1.25 | 1.78 | 3.32 |
| *Event Instance Counts* | | | |
| Train | 7,329 | 3,241 | 4,442 |
| Dev | 924 | 345 | - |
| Test | 871 | 365 | 2,200 |

We evaluate on three document-level EAE benchmarks: RAMS (9,124 events from news, 39 types/65 roles) Ebner et al. (2020), WikiEvents (246 documents, 50 types/59 roles) Li et al. (2021), and biomedical-domain MLEE (23 event types) Pyysalo et al. (2012). All follow standard preprocessing Ma et al. (2022); Xu et al. (2022) with two adaptations: 1) For RAMS, we merge events within the same document while retaining sentence segmentation; 2) For MLEE without validation set, we follow He et al. (2023) to use training data for validation. Dataset statistics are in Table 6.

For enhanced training, we incorporate two sentence-level datasets: ACE (English subset) Doddington et al. (2004) for news domain alignment, and multi-domain GENEVA Parekh et al. (2023). Both are preprocessed following Wadden et al. (2019); Hsu et al. (2023b). This combination leverages ACE's event extraction prominence and GENEVA's domain diversity.

All models are evaluated on test sets with standard metrics. Domain distribution: RAMS/WikiEvents (news), MLEE (biomedical). Sentence counts per document vary in WikiEvents, while RAMS uses fixed 5-sentence documents. Biomedical specificity makes MLEE particularly challenging for Multi Events.Detailed statistics of the above datasets are listed in Table 6.

We included the ACE and GENEVA datasets in the SFT mainly because: the basic multiple dataset SFT ,CsLLM (Doc), primarily validates the cross-dataset synergistic effects in document-level EAE. In contrast, the extended strategy incorporating additional datasets carries dual groundbreaking significance:

1. It verifies the feasibility of cross-data-level transfer that integrating sentence-level data into document-level task training. Empirical studies demonstrate that annotation data of different granularities can facilitate positive knowledge transfer.

2. It systematically demonstrates the effectiveness of cross-domain generalization mechanisms. The inclusion of ACE (news domain) and GENEVA (general domain) proves that, even when the

target domain lacks sufficient training data, auxiliary datasets from heterogeneous domains or with heterogeneous structures (e.g., sentence-level) can significantly enhance model performance. This finding provides crucial insights for researchers facing data scarcity in practical applications: strict constraints on the domain relevance and structural consistency of auxiliary data are unnecessary, as performance gains can be achieved through our multi-source fusion framework.

## C  IMPLEMENTATION

We used PyTorch and a single NVIDIA A40 Tensor Core GPU with 45GB to train all models and reproduce experiments of other models. We fine-tune this LLM using LoRA Hu et al. (2021) with a rank $r = 8$, scaling factor $\alpha = 32$, and a dropout rate of 0.1. We use the AdamW optimizer, with an initial learning rate set to $5 \times 10^{-5}$. We used BART Lewis et al. (2020) as the backbone for CsEAE, and the number of parameters is around 406 millions. During model training the learning rate was set to 2e-5. We used version 2.0.1 of Torch to build CsEAE. The training steps is 10000 and batch size is 4 for all datasets.

## D  EVALUATION METRICS

Following the same evaluation metrics as in prior works Li et al. (2021); Hsu et al. (2022); Ma et al. (2022); Yang et al. (2023); He et al. (2023); Xu et al. (2022) for all datasets, we used the Arg-I F1 score and Arg-C F1 score to evaluate the model's performance on Argument Identification and Argument Classification tasks, respectively.

We considered TP as true positives, FN as false negatives, and FP as false positives. Recall (R) can be calculated using TP / (TP + FP), and precision (P) can be calculated using TP / (TP + FN). The F1 score combines both recall and precision, defined as F1 = 2 * P * R / (P + R).

• Arg-I: an argument is correctly identified from event mention.

• Arg-C: an argument is correctly classified if its offset and the role's label both match the ground truth.

Since the Arg-C score reflects whether the model extracts the correct arguments and associates them with the appropriate roles to generate the correct structured events, the EAE task places more emphasis on the Arg-C F1 score.

## E  STRUCTURE-AWARE INTERACTION

To analyze the effectiveness of the model in performing extraction centered around the sentence containing the trigger word, we conducted an analysis on RAMS, which has the highest number of cross-sentence arguments. We defined the distance D between a role and the trigger as the maximum argument distance among all arguments for that role. When the trigger and the maximum argument are in the same sentence, D=0; when they are not, D≠0. In the Table 7, CsEAE achieved a 3.23% improvement in the Arg-C metric compared to PAIE when D=0. This improvement significantly contributed to CsEAE's overall lead over PAIE in all datasets. The substantial improvement at D=0 also demonstrates that the model's approach of centering the document structure around the trigger's sentence effectively helps focus attention on the core content of the sentence, reducing the distraction from redundant information. Furthermore, CsEAE also excelled when D≠0, achieving a 3.77% improvement over TabEAE.

Table 7: Performance on cross-sentence arguments.

| Model | RAMS (Arg-C F1) | | |
|---|---|---|---|
| | D=0 | D≠0 | Overall |
| PAIE | 58.7 | 35.3 | 51.0 |
| TabEAE | 61.2 | 31.8 | 51.4 |
| **CsEAE** | 61.9 | 35.5 | 53.1 |

## F    IN-CONTEXT LEARNING WITH LLMS

As shown in the Table 3, in the ICL setting, the Open-AI series models demonstrated superior performance compared to the Open-resource models. Notably, instruct-type models have shown relatively poor performance during ICL. However, after fine-tuning, they outperformed base models on some datasets.

Additionally, we can see that Llama-Instruct scores 0 on all datasets. We analyzed that this result is likely due to the fact that instruction-based models tend to generate safe and concise answers that align with human feedback, which may suppress in-depth reasoning for complex problems. Lastly, our instructions are more complex compared to general instructions, and EAE itself is a complex extraction task. Therefore, without fine-tuning, the model's performance is not satisfactory.

## G    GENERALIZATION OF LLMS

To analyze the generalization challenges of LLMs in broader domains and their applicability in real-world scenarios, we conducted extensive experiments on the GENEVA dataset, which includes 115 event types and 220 distinct roles across general-domain, sentence-level data. The experimental results are presented in the Table 8. Surprisingly, unlike in domain-specific document-level datasets, multiple datasets SFT does not enhance model performance on GENEVA. However, incorporating co-occurrence-aware and structure-aware interactions into the prompt improves the model's performance on document-level datasets, allowing for better extraction on GENEVA. This indicates that the model learns to capture co-occurrence-aware and structure-aware information from the three document-level datasets, such that, even though sentence-level datasets cannot directly embed structure-aware information in prompt construction, the model can leverage what it learned from document-level data to assist in extraction. Additionally, it becomes evident that LLMs do not perform well on general-domain datasets like GENEVA. Its best performance, an Arg-C score of 64.71, falls short compared to best results of smaller model  Huang et al. (2024). We attribute this to the fact that many event types in GENEVA are quite similar, and fine-tuning an 8B-parameter model using prompt + LoRA struggles to discern numerous labels and their subtle interactions during extraction Ma et al. (2023).

Table 8: Overall performance of LLMs on GENEVA.

| GENEVA | | |
|---|---|---|
| **Model** | **Arg-I** | **Arg-C** |
| *In-Context Learning (ICL)* | | |
| GPT-3.5 | 33.07 | 27.97 |
| GPT4o-mini | 35.17 | 31.06 |
| GPT4o | 42.98 | 39.55 |
| Llama3 | 4.70 | 3.61 |
| Llama3-Instruct | 0.35 | 0.29 |
| *Supervised Fine-tuning* | | |
| Llama3 | 28.98 | 27.88 |
| Llama3-Instruct | 66.07 | 62.42 |
| **CsLLMs (ALL)** | 67.99 | 64.71 |

## H    ZERO-SHOT TRANSFER CAPABILITY

Additionally, we conducted further investigations into zero-shot transfer capabilities, as summarized in Table 9.

First, we validated the one-shot and zero-shot capabilities of the models under GPT4o and GPT4o-mini. Using the same datasets as before, within the ICL framework, both GPT4o and GPT4o-mini exhibited performance degradation in the zero-shot setting compared to their one-shot counterparts.

Unlike the one-shot scenario, in the zero-shot setting, even after SFT, the models failed to surpass the performance of GPT4o when faced with general datasets. This indicates that without the dual role of demonstration examples in the one-shot setting: (1) serving as task paradigm examples to guide

the model in understanding extraction logic, and (2) acting as domain feature anchors to help the model establish boundaries between event types and datasets—the fine-tuned models were unable to outperform the larger model like GPT4o.

Table 9: Zero-shot transfer experiments on GENEVA.

| GENEVA | | |
|---|---|---|
| Model | Arg-I | Arg-C |
| **One-shot** | | |
| GPT4o | 42.98 | 39.55 |
| GPT4o-mini | 35.17 | 31.06 |
| **Zero-shot** | | |
| GPT4o | 38.73 | 35.69 |
| GPT4o-mini | 29.99 | 25.95 |
| **CsLLM (Doc)** | 23.09 | 20.72 |

## I  CASE STUDY

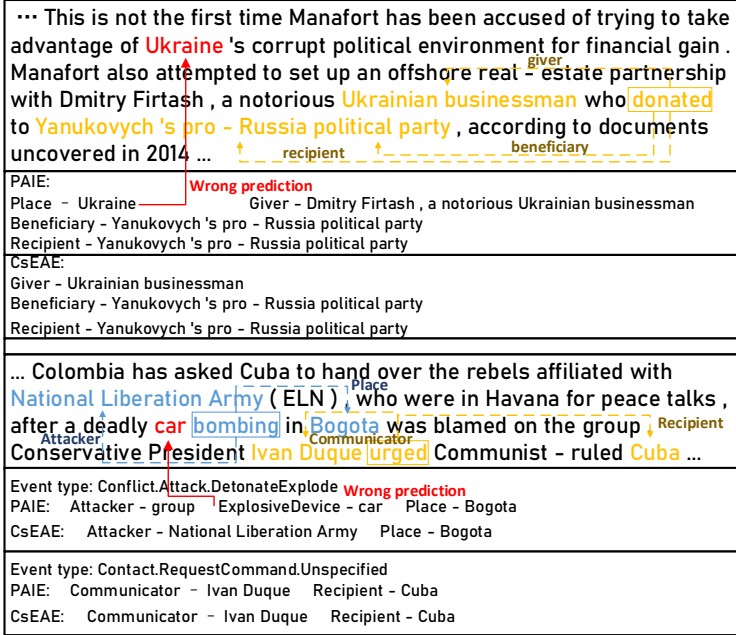

Figure 6: Two test cases from RAMS and WikiEvents.

In the first case of Figure 6, PAIE incorrectly predicts the token span *Ukraine* from the previous sentence of the correct argument's sentence as the argument for the role Place. In contrast, CsEAE avoids this error by leveraging the structure-aware module to enhance the model's attention on the sentence containing the trigger, thereby mitigating interference from redundant information in other sentences. In the second example, PAIE erroneously identifies *car* as the argument for ExplosiveDevice, while CsEAE avoids this mistake by incorporating information from co-occurring events and utilizing strong causal relationships across multiple events.

