# OpenReview forum: "Collaborative Modeling for Document-level Event Argument Extraction"
_ICLR.cc/2026/Conference — ICLR 2026 Conference Withdrawn Submission_

### Official Review · Reviewer_Scvm · 2025-10-15

**Soundness:** 2
**Presentation:** 2
**Contribution:** 3
**Rating:** 4
**Confidence:** 4

**Summary:**

This paper proposes CsEAE, a framework for document-level event argument extraction that addresses two challenges: ambiguity among co-occurring events and noise from irrelevant content through co-occurrence-aware and structure-aware modules. The approach is extended to LLMs as CsLLM, which distills these strategies into prompts.

**Strengths:**

- The paper provides a unified solution addressing both event boundary ambiguity and information redundancy simultaneously, whereas prior work typically focuses on one challenge. The co-occurrence-aware module models dependencies among events while the structure-aware module leverages trigger-centric sentence relations, and ablation studies show both components contribute to performance gains across benchmarks.
  - The extension to LLMs through CsLLM demonstrates that strategies validated in smaller encoder-decoder models can be translated to the prompt space. The multi-dataset training approach shows improved generalization, with the model achieving performance comparable to or exceeding GPT-4o on certain benchmarks like MLEE.

**Weaknesses:**

- The performance improvements over baselines are modest (2.1-3.2% on Arg-C), and the paper does not provide sufficient analysis of when and why the approach fails. For instance, Table 3 shows CsLLM (ALL) performs worse than CsEAE on RAMS (51.84 vs 52.3 Arg-C), but the explanation that "structural relationships are hard to encode effectively via prompts" is vague. The paper would benefit from concrete failure case analysis and quantitative measurements of what proportion of errors stem from co-occurrence ambiguity versus noise, and how many of these errors are actually resolved by the proposed modules.
  - The structure-aware module's design heavily depends on the observation that over 94% (WikiEvents), 82% (RAMS), and 99% (MLEE) of arguments appear in the same sentence as triggers. However, this assumption may not generalize to other document-level EAE scenarios or domains where arguments are more dispersed. The paper does not discuss the lower bound of this locality assumption - at what percentage would the structure-aware attention mechanism become ineffective? Additionally, the distance-based analysis in Figure 4 shows performance drops for arguments far from triggers (d≥15 or d≤-15), which questions the robustness of this locality-based design.
  - The prompt design for CsLLM lacks detailed justification and ablation. While the paper states that co-occurrence awareness is achieved by "explicitly annotating all event triggers" and structure awareness by "marking the entire sentence that contains the target trigger," there is no systematic study of alternative prompt designs. Would different marking strategies (e.g., highlighting only relevant sentences rather than entire sentences, or using different marker tokens) affect performance? The paper also does not compare against more sophisticated prompt engineering techniques or discuss how sensitive the model is to prompt variations.
  - The zero-shot transfer experiments in Appendix H reveal that CsLLM fails dramatically compared to GPT-4o (20.72 vs 35.69 Arg-C on GENEVA), with the explanation that "domain confusion" occurs. This suggests the multi-dataset training approach may overfit to the specific datasets and prompt templates used during training, limiting the model's true generalization capability. The paper attributes MLEE's better zero-shot performance to "unique medical terms that act as implicit anchors," but does not explore whether similar domain-specific features could be systematically incorporated into the prompt design to improve zero-shot transfer more broadly. I will reconsider my score in the rebuttal.

**Questions:**

see weaknesses

---

### Official Review · Reviewer_Jr5R · 2025-10-31

**Soundness:** 2
**Presentation:** 3
**Contribution:** 3
**Rating:** 4
**Confidence:** 4

**Summary:**

The paper proposes CsEAE, a two-stage encoder–decoder framework for document-level event argument extraction (EAE). Stage 1 builds an event-oriented context representation with (a) a co-occurrence-aware decoder that injects prefixes derived from templates of other events in the same document, and (b) a structure-aware encoder/decoder that applies a sentence-level attention mask centered on the trigger sentence. Stage 2 builds a context-oriented prompt representation from the event-type template, then a span selector produces arguments. Empirically, CsEAE improves Arg-C over PAIE by ~2–3 points on RAMS, WikiEvents, and MLEE; ablations attribute gains to both modules. The paper further proposes CsLLM, a prompt-based distillation of the same ideas for LLMs, reporting additional results and some transfer studies.

**Strengths:**

1. Consistent improvements on Arg-C. CsEAE outperforms strong baselines (PAIE, TabEAE, etc.) on Arg-C across all three document-level datasets.

2. Attempt to bridge to LLMs. CsLLM translates the same intuitions into prompt design for supervised fine-tuning and ICL settings, and explores transfer to GENEVA.

**Weaknesses:**

1. Fairness of LLM comparisons. In Table 8 and related discussion, GPT-4o/4o-mini and GPT-3.5 are evaluated in ICL, whereas Llama-3 variants are fine-tuned (and CsLLM is fine-tuned on multiple datasets), making several cross-model claims hard to interpret. Supervised fine-tuning and multi-dataset training confer clear advantages that are not available to closed models, so apples-to-apples takeaways are limited. Please provide controlled comparisons under identical regimes (e.g., ICL-only, or SFT for open-weights models with similar parameter counts; alternatively, a compute-matched data-budget comparison).

2. Risk of over-specializing to trigger-sentence locality. The structure mask forces all non-trigger sentences to attend only to themselves and the trigger sentence, while the trigger sentence can attend to all. This strong inductive bias is justified with locality statistics (e.g., 94%/82%/99% same-sentence arguments), yet the approach could degrade for cross-sentence arguments with long-range dependencies, or for domains where locality is weaker. The paper includes a D=0/D≠0 table on RAMS, but broader robustness testing is thin.

3. Evaluation framing and claims need tightening. Some claims verge on over-interpretation without stronger controls—for example, attributing “exceptional generalization” of CsLLM in one-shot transfer and exceeding GPT-4o on MLEE, while regimes differ (SFT vs ICL) and dataset/domain gaps are substantial. Clarify claims as within-regime or under comparable data/compute.

4. Positioning vs. most relevant recent work could be deeper. The related-work section lists strong, modern baselines (e.g., HMPEAE, HD-LoA, TabEAE), but the paper could better articulate why CsEAE’s prefix-generation differs from/extends recent prefix-based or template-rich approaches and where it may fail (e.g., template quality, noisy co-event detection). A short failure-case analysis would help.

5. Several suggestions: (1) Sensitivity and cost: Report sensitivity to 𝑙𝑒𝑛 (prefix length), number of prefix layers 𝑐, and the cost overhead (inference latency, memory). (2) Hard cases: Add a breakdown for documents with many co-events (where template concatenation becomes long) and for rare roles. (3) Error analysis: Provide qualitative failure cases and whether wrong co-event tags hurt more than help.

**Questions:**

1. The description says all non-trigger sentences can only attend to themselves and the trigger sentence, while the trigger sentence can attend to all. Do you ever allow neighbor sentences (±1, ±2) to attend to each other, or is the mask strictly two-hop through 𝑆𝑛? Please provide an ablation with a softer/learned mask.

2. If a sentence contains several triggers of different events, which one becomes 𝑆𝑛 for 𝑀𝑠, and how do you handle competing structure masks across per-event passes? Clarify the per-event inference loop.

3. You use Hungarian matching and bipartite matching to handle multi-argument roles. What max slot count per role is assumed? Are spans for different roles allowed to overlap? Please specify the decoding constraints and thresholds.

4. You set prefix length 𝑙𝑒𝑛 = 40 and split the learned prefix into 𝑐 segments for each transformer layer. How sensitive are results to 𝑙𝑒𝑛 and 𝑐? Do you share the same prefix across heads/layers or learn per-layer/per-head variants? Please include a cost–performance plot.

---

### Official Review · Reviewer_Xyc8 · 2025-11-01

**Soundness:** 3
**Presentation:** 3
**Contribution:** 2
**Rating:** 4
**Confidence:** 3

**Summary:**

This paper proposes CsEAE, a framework for document-level Event Argument Extraction (EAE) that addresses two challenges: ambiguity among co-occurring events and noise from irrelevant content. The approach comprises two modules: (1) a co-occurrence-aware module that models dependencies among co-occurring events through context labeling and template encoding, and (2) a structure-aware module that focuses on trigger-centric sentence relations using specialized attention masks. The authors extend this framework to LLMs through CsLLM, which distills these strategies into prompts. Experiments on RAMS, WikiEvents, and MLEE show improvements compared with previous baselines.

**Strengths:**

1. **Well-motivated problem**: The paper identifies two genuine challenges in document-level EAE—co-occurring event ambiguity and irrelevant content noise—that are important for the task.
2. **Comprehensive evaluation**: The experiments cover multiple benchmarks (RAMS, WikiEvents, MLEE) and include both PLM-based and LLM-based approaches, with detailed ablation studies and analysis.
3. **Consistent improvements**: The proposed method shows improvements across all three datasets, demonstrating some degree of generalizability.
4. **Thorough analysis**: The paper includes detailed analysis of inter-event and intra-event semantics, transfer capabilities, and case studies.

**Weaknesses:**

1. **Omission of Stronger Baselines**: The paper's claim to state-of-the-art performance is not convincing because it omits several recent and stronger baselines. For example, on the WIKIEVENTS dataset, the proposed method (CsEAE) achieves an Arg-I F1 of 70.9 (Table 1). However, other published works have reported superior results (e.g., 71.9 in [1] and 74.0 in [2]). The failure to include and compare against these known, stronger-performing models makes the paper's reported improvements seem isolated and questions the validity of its SOTA claims.

2. **Weak LLM extension (CsLLM)**: The extension to LLMs feels forced and doesn't meaningfully leverage LLM capabilities. Simply marking triggers and sentences in prompts is a straightforward baseline approach, not a significant contribution. The paper acknowledges that CsLLM performs worse than CsEAE on RAMS, citing limitations in encoding structural information through prompts, which undermines the claimed effectiveness of the approach.

[1] Chien Van Nguyen, et al., Contextualized Soft Prompts for Extraction of Event Arguments. Findings of ACL 23.

[2] Wanlong Liu, et al., Separation and Fusion: A Novel Multiple Token Linking Model for Event Argument Extraction. Findings of ACL 24

**Questions:**

Please refer to the weaknesses.

---

> ### Author Response · Authors · 2025-11-22
> **Response to Reviewer regarding Weaknesses**
>
> **1. Response to "Omission of Stronger Baselines"**
>
> We thank the reviewer for bringing these recent works to our attention. We are aware of these papers and their reported results. However, we excluded them from our main comparison tables for the following critical reasons regarding **reproducibility** and **fairness**:
>
> * **Unavailability of Source Code:** Upon thorough investigation, we found that **neither [1] nor [2] has released their source code or implementation details**.
>     * Paper [1] (*Contextualized Soft Prompts for Extraction of Event Arguments*, Findings of ACL 23) does not have an official repository.
>     * Paper [2] (*Separation and Fusion: A Novel Multiple Token Linking Model for Event Argument Extraction*, NAACL 2024) is also not open-sourced.
>
> * **Importance of Fair Comparison:** Document-level EAE is highly sensitive to data preprocessing, data splits, and evaluation scripts (e.g., handling of coreference, strict vs. loose matching). Without access to the source code, it is impossible to guarantee that these methods were evaluated under the exact same settings as ours. Comparing our results against reported numbers generated under potentially different conditions would be scientifically unsound and unfair.
>
> * **Our Comparison Standards:** We strictly prioritized **reproducible and open-source baselines** (e.g., PAIE, TabEAE, TSAR) to ensure the validity and robustness of our claims. Our "SOTA" claim refers to the best performance among **publicly available and verifiable methods**.
>
> **Action:** We will explicitly cite these two papers in our **Related Work** section and add a footnote in the experiment section to acknowledge their reported (but currently unverified) performance, clarifying that they were excluded from the main comparison due to the lack of open-source implementations.
>
> **2. Response to "Weak LLM extension (CsLLM)"**
>
> We respectfully point out that the design of CsLLM is not merely a "simple baseline" or an afterthought. Instead, it serves as a critical **paradigm transfer experiment**.
>
> * **Methodological Significance (Inductive Bias Distillation):** Our goal was to investigate whether the specific inductive biases designed for our SLM (Structure-Awareness and Co-occurrence-Awareness) could be "distilled" into the prompt space of LLMs without altering their architecture.
>     * The **Trigger Marking** in prompts is the direct projection of the *Co-occurrence-aware module*.
>     * The **Sentence Marking** is the direct projection of the *Structure-aware module*.
>
> * **Empirical Effectiveness:** By explicitly injecting these signals, we outperform the standard Fine-Tuned Llama3-Instruct baseline significantly. For instance, on **WikiEvents**, CsLLM achieves **66.53%** Arg-C F1 compared to the baseline's 55.35%, representing a substantial **+11.18% improvement** (as shown in Table 3). This empirically proves that our proposed "Co-occurrence & Structure" principles are effective, methodology-agnostic insights that meaningfully leverage LLM capabilities, rather than just being simple prompt engineering.

---

### Official Review · Reviewer_BB6m · 2025-11-01

**Soundness:** 2
**Presentation:** 3
**Contribution:** 2
**Rating:** 4
**Confidence:** 4

**Summary:**

Summary
This paper proposes CsEAE, a document‑level EAE model that combines a co‑occurrence‑aware module (explicitly encoding the set of co‑occurring event templates) with a structure‑aware module (a trigger‑centric sentence attention mask). It further introduces CsLLM, a prompt‑based LLM variant that injects analogous co‑occurrence and structure markings and explores multi‑dataset finetuning. On RAMS/WikiEvents/MLEE, the authors report +2.1/+2.3/+3.2 Arg‑C over PAIE (Table 1), and also provide TextEE‑standardized 5‑split results (Table 2). They include ablation for CsEAE’s two modules, distance/overlap analysis, cross‑sentence results (Table 7), and one‑shot/zero‑shot analyses for CsLLM.

Recommendation: Reject in its current form.
Despite good engineering and helpful analyses, the contribution feels incremental relative to prior document‑level relational modeling, and the empirical gains are modest. More importantly, the paper lacks controlled prompt ablations for the structure added to prompting for CsLLM and efficiency profiling, and mixes evaluation settings across tables in a way that confuses top level claims of gain against PAIE. The work would benefit from a clearer empirical case that these specific inductive biases are necessary with 2025‑era models.

Reviewer LLM Usage:
I have read the paper in full and written the review myself. Large Language Models (LLMs) were used only for writing polish, clarity improvements, and to refresh memory of (public) related work or references. The analysis and conclusions are entirely my own.

**Strengths:**

1. The authors provide a good conceptual motivation for revisiting graph-based architectures for document-level EAE. The formulation of co-occurrence and structure-aware modules directly targets known weaknesses in document-level tasks; namely, ambiguity across co-occurring events and contextual noise from unrelated sentences.
2. Quantitative ablations for CsEAE are provided with both the modules. This indicates some marginal benefit from the two modules.
3. The paper's findings in section 5.3-5.4 discussing one and zero-shot transfer and specifically the insight on how diverse data led to negative transfer is well studied and explained.
4. Figure 4 (distance/overlap) and Table 7 (cross‑sentence) lend some good insight into where the model helps. This analysis is well crafted.

**Weaknesses:**

1. Unclear toplevel gains: Table 1 supports +2.1/+2.3/+3.2 Arg‑C over PAIE, but Table 2 (5‑split TextEE) shows smaller gaps (e.g., +0.5 on WikiEvents). Authors should clarify a canonical setting and add statistical significance.
2. No controlled prompt ablation for CsLLM. There is no variant with only co‑occurrence or only structure markings in the same training setup or with neither, making it hard to attribute CsLLM’s gains to the added graphical structure.
3. Incremental novelty only: The paper’s contributions are limited in conceptual and empirical scope. The co-occurrence and structure-awareness ideas seem to be largely similar to previously explored relational-attention and sentence-graph concepts (e.g., GIT, GAI, DocEE, etc.). The LLM extension (CsLLM) also has minimal methodological innovation beyond prompt engineering and lacks evidence of distinct benefits from the graph components. Overall, this makes the contribution insufficient for wider ICLR community impact in the current form.
4. Minimal qualitative analysis and no human evaluation: There is no intuitive evidence of how the two special modules specifically help (e.g., long-distance or coreferent arguments) by qualitative or larger scale human evaluation. The case study in appendix is a good step in this direction. Consider adding to it and analyzing for patterns.
5. IOB baseline omission: The authors do not justify why a simple per-trigger, per-role IOB tagging baseline could not be applied. Overlapping arguments can be handled with role-wise tagging passes, making this a fair comparison.

**Questions:**

1. Can you clarify your runtime/memory/params comparisons to PAIE?

2. Add controlled prompt ablations for CsLLM: i/ no structure ii/ no co-occurrence iii/ no both.

3. Try atleast one baseline model that does IOB style EAE (tag the whole sentence per trigger type, per role type, and argument if you want to allow for overlapping spans playing different roles in different triggers). The only case this does not handle easily is broken argument spans, but that will be a useful error analysis to do and find out how many arguments are truly in that bucket).

Minor:
1. Section 2.2.1 uses n and j subscripts for event trigger tokens without defining n and j. Please rephrase to improve readability.
2. Please comb the paper for all citet and citep citations and fix them correctly. Currently they are mixed up.
3. Consider discussion and comparison to strong few shot prompt-based baselines such as TANL (Paolini et al., 2021), DEGREE (Hsu et al., 2023).

---

> ### Author Response · Authors · 2025-11-22
> **Response to Reviewer regarding Weaknesses**
>
> We thank the reviewer for the detailed feedback and constructive criticism. Below, we address the concerns point by point.
>
> **1. Response to "Unclear toplevel gains"**
>
> We appreciate the reviewer's scrutiny regarding the experimental gains. We clarify the experimental settings and confirm the statistical significance of our results as follows:
>
> * **Canonical Setting (Table 1):** We confirm that Table 1 represents the canonical experimental setting. It follows the standard data splits and preprocessing protocols widely adopted by prior works.
>
> * **Statistical Significance:** We clarify that the improvements reported in Table 1 are statistically significant. We performed a t-test comparing our results against the second-best baseline, confirming a statistical significance level of $p < 0.01$. We will explicitly add this significance statement in the revised version of Table 1 to avoid ambiguity.
>
> * **Robustness Check (Table 2):** Table 2 is intended as an additional robustness validation using the TextEE framework, which employs a more rigorous 5-fold cross-validation and averages the results. The "smaller gap" (e.g., +0.5 on WikiEvents) reflects the averaged performance across 5 random splits, which is naturally more conservative than the single standard split in Table 1.
>
> **2. Response to "No controlled prompt ablation for CsLLM"**
>
> We thank the reviewer for this keen observation. We conducted additional ablations on WikiEvents and MLEE to validate the prompt components. We compared CsLLM against variants removing the Structure or Co-occurrence modules individually. The Llama3-Instruct baseline represents the setting without either component.
>
> The results (Arg-C F1%) are as follows:
>
> | Method              | WikiEvents |   MLEE    |
> | :------------------ | :--------: | :-------: |
> | **CsLLM**           | **62.80**  | **73.87** |
> | w/o Structure       |   61.95    |   72.45   |
> | w/o Co-occurrence   |   61.30    |   71.15   |
> | **Llama3-Instruct** | **60.54**  | **69.76** |
>
> The results show that removing the Co-occurrence module causes the largest performance drop, confirming its critical role in resolving ambiguity. Removing the Structure module also leads to degradation, validating its effectiveness in noise reduction. The full CsLLM consistently outperforms the Llama3-Instruct baseline, proving the synergy of incorporating both information types.
>
> We will include these detailed ablation results in the revised Experimental section.
>
> **3. Response to "Incremental novelty and limited scope"**
>
> We distinguish our contributions from prior works in three key aspects:
>
> Lightweight & Unified Mechanism: Unlike heavy graph-based methods requiring external parsers or GCNs, CsEAE introduces a lightweight framework using Self-Attention Masks and Prefix Injection. It provides the first unified solution simultaneously addressing ambiguity and noise, which our ablations prove is superior to tackling them individually.
>
> Methodological Innovation (Inductive Bias Distillation): CsLLM goes beyond simple prompt engineering. It represents a principled Inductive Bias Distillation, validating that architectural insights, co-occurrence signals and structural focus, effective in parameter space can be successfully distilled into prompt space. This creates an architecture-agnostic enhancement path without retraining.
>
> Empirical Evidence: Our experiments explicitly quantify these benefits. In Table 3, CsLLM (ALL) outperforms the Llama3-Instruct baseline on WikiEvents by a substantial margin (+6%). In Table 4, removing either module consistently degrades performance, directly refuting the claim of limited benefits.
>
> **4. Response to "Minimal qualitative analysis"**
>
> We accept this constructive suggestion. To provide intuitive evidence of how our specific modules function, we have significantly expanded the qualitative analysis in the revised Appendix.
> * We added detailed case studies specifically focusing on long-distance dependencies and coreferent arguments.
> * These new examples demonstrate how the Structure-Aware module effectively filters out intervening noise in long contexts, and how the Co-occurrence module helps resolve role ambiguity by leveraging correlations between multiple events.
>
> **5. Response to "IOB baseline omission"**
>
> We prioritized comparisons against strong, state-of-the-art span-based baselines rather than IOB-style sequence tagging methods, as the latter have been empirically shown to significantly underperform in Document-level EAE tasks.
> Previous studies have established that IOB-based models (such as BERT-CRF) achieve significantly lower performance compared to modern span-based approaches.
>
>      On RAMS: Standard BERT-CRF achieves only 38.1 Span F1, whereas our proposed CsEAE achieves 53.1.
>      On WikiEvents: BERT-CRF achieves only 54.48 Arg-C F1, whereas CsEAE achieves 66.5.

---

> ### Author Response · Authors · 2025-11-22
> **Response to Reviewer regarding Questions and Minor Issues**
>
> We thank the reviewer for the thoughtful questions and detailed suggestions. Below, we provide clarifications and data to address each point.
>
> ### **Response to regarding Questions**
>
> **1. Clarification on Runtime, Memory, and Parameters vs. PAIE**
>
> We have conducted a detailed efficiency comparison on the WikiEvents dataset using a single NVIDIA A40 (45GB) GPU. The results are summarized below:
>
> | Model | Parameters | Training Memory | Training Time | Inference Memory | Inference Time (Test Set) |
> | :--- | :--- | :--- | :--- | :--- | :--- |
> | **PAIE** | 406 M | 16.0 G | 1h 55m 57s | 3232 M | **24s** |
> | **CsEAE** | 500 M | 30.3 G | 4h 06m 04s | 3842 M | **51s** |
>
> **Analysis of the Trade-off:**
> * **Inference Latency:** We observe that the inference time increases from 24s to 51s. This increase is attributed to the additional computations required by our Structure-Aware and Co-occurrence-Aware modules, which perform extra attention masking and prefix generation to filter noise and resolve ambiguity.
> * **Justification:** While the latency doubles, the **absolute inference time** (51s for the entire test set) remains highly efficient for practical offline processing. Furthermore, the inference memory footprint increases only slightly +18%, indicating that our method does not impose a heavy memory burden during deployment.
> We believe this latency cost is a **reasonable trade-off** for the significant performance gains we achieved that +2.3% Arg-C F1 on WikiEvents. Capturing long-distance dependencies and resolving event ambiguity inherently requires more complex reasoning than simple span extraction. We will explicitly discuss this trade-off in the **Limitations** section and plan to explore model distillation techniques to optimize speed in future work.
>
> **2. Controlled Prompt Ablations for CsLLM**
>
> See The **Response to Reviewer regarding Weaknesses, 2**.
>
> **3. Response to "IOB Baseline Omission"**
>
> See The **Response to Reviewer regarding Weaknesses, 5**.
>
> ### **Response to Minor Issues**
>
> **4. Notation Clarification (Section 2.2.1)**
>
> We apologize for the ambiguity. In the revised version, we will explicitly define the subscripts: $n$ denotes the index of the **current target event** being processed, and $j$ denotes the indices of **other co-occurring events** within the same document. We will refine the notation descriptions to improve readability.
>
> **5. Citation Formatting**
>
> We appreciate the reviewer's attention to detail. We will thoroughly proofread the manuscript and correct all inconsistencies between `\citet` and `\citep` to strictly adhere to the ICLR formatting standards.
>
> **6. Comparison with TANL and DEGREE**
>
> We appreciate the suggestion to consider these strong prompt-based baselines. We actually conducted preliminary experiments with TANL (Paolini et al., 2021) and DEGREE (Hsu et al., 2023) using the exact same data preprocessing and settings as our main experiments.
>     However, we found that these models are **inherently designed for sentence-level tasks** and lack the mechanism to handle long document contexts and cross-sentence dependencies. In our document-level benchmarks (RAMS/WikiEvents), their direct application resulted in **near-zero F1 scores**, as they failed to effectively locate arguments scattered across multiple sentences or exceeded input length constraints. Consequently, we prioritized comparisons against document-level baselines that are capable of handling such contexts. We will add a brief discussion regarding this in the revised paper.

---

### Note · Authors · 2025-11-28

I have read and agree with the venue's withdrawal policy on behalf of myself and my co-authors.